# Effects of Supplemental Light Spectra on the Composition, Production and Antimicrobial Activity of *Ocimum basilicum* L. Essential Oil

**DOI:** 10.3390/molecules27175599

**Published:** 2022-08-31

**Authors:** Ha Thi Thu Chu, Thi Nghiem Vu, Thuy Thi Thu Dinh, Phat Tien Do, Ha Hoang Chu, Tran Quoc Tien, Quang Cong Tong, Manh Hieu Nguyen, Quyen Thi Ha, William N. Setzer

**Affiliations:** 1Institute of Ecology and Biological Resources, Vietnam Academy of Science and Technology (VAST), 18 Hoang Quoc Viet, Cau Giay, Ha Noi 10072, Vietnam; 2Graduate University of Science and Technology, VAST, 18 Hoang Quoc Viet, Ha Noi 10072, Vietnam; 3Institute of Materials Science, VAST, 18 Hoang Quoc Viet, Ha Noi 10072, Vietnam; 4Institute of Natural Product Chemistry, VAST, 18 Hoang Quoc Viet, Ha Noi 10072, Vietnam; 5Institute of Biotechnology, VAST, 18 Hoang Quoc Viet, Ha Noi 10072, Vietnam; 6Faculty of Agricultural Technology, VNU University of Engineering and Technology, Vietnam National University Hanoi, 144 Xuan Thuy, Ha Noi 10053, Vietnam; 7Aromatic Plant Research Center, 230 N 1200 E, Suite 100, Lehi, UT 84043, USA; 8Department of Chemistry, University of Alabama in Huntsville, Huntsville, AL 35899, USA

**Keywords:** antimicrobial activity, basil, essential oil composition, light spectra, methyl chavicol, *Ocimum basilicum*

## Abstract

This study was performed to investigate the effects of different supplemental light spectra and doses (duration and illuminance) on the essential oil of basil (*Ocimum basilicum* L.) cultivated in the net-house in Vietnam during four months. Ten samples of basil aerial parts were hydrodistilled to obtain essential oils which had the average yields from 0.88 to 1.30% (*v*/*w*, dry). The oils analyzed using GC-FID and GC-MS showed that the main component was methyl chavicol (87.4–90.6%) with the highest values found in the oils of basil under lighting conditions of 6 h/day and 150–200 µmol·m^−2^·s^−1^. Additional lighting conditions caused the significant differences (*p* < 0.001) in basil biomass and oil production with the highest values found in the oils of basil under two conditions of (1) 71% Red: 20% Blue: 9.0% UVA in at 120 μmol·m^−2^·s^−1^ in 6 h/day and (2) 43.5% Red: 43.5% Blue: 8.0% Green: 5.0% Far-Red at 100 μmol·m^−2^·s^−1^ in 6 h/day. The oils of basil in some formulas showed weak inhibitory effects on only the *Bacillus subtilis* strain. Different light spectra affect the biomass and essential oil production of basil, as well as the concentrations of the major components in the oil.

## 1. Introduction

Basil (*Ocimum basilicum* L.) is an herb, probably native to the tropical and subtropical regions of India, well known for its aromatic and medicinal properties. This herb contains a distinctive essential oil and is used in both fresh and dried culinary dishes, providing a variety of positive health benefits when consumed. Some of the reports demonstrated that basil has pharmacological effects such as anti-inflammatory [1,2,3], antioxidant [4,5], and bronchodilatory properties [6]. In addition, basil has strong immunomodulatory, antibacterial, antimutagenic, and chemopreventive effects [7]. Basil was used in the treatment of obstructive lung diseases, e.g., chronic obstructive pulmonary disease (COPD), asthma, and other respiratory disorders such as bronchitis, aspergillosis, tuberculosis, and lung cancer [8]. Due to the unique properties of smell, taste, and biological activities, Basil is highly appreciated in the world market with a steady increase of 1.3% cumulative average growth rate, which is projected to show an increase from 57 million USD in 2020 to 62 million USD in 2026 [9].

The composition as well as production of essential oils and bioactive compounds in plants are influenced by genetic and environmental factors. Among the environmental factors, light not only provides vital energy for plant growth and development but also acts as an external signal to trigger plant-specific responses [10]. It influences the accumulation of phytochemicals such as flavonoids, ascorbic acid, carotenoids, tocopherols, enzyme antioxidants, and the biosynthesis of essential oils. As chlorophyll pigments absorb mainly in the red (R) and blue (B) regions of light, these wavelengths are the major energy sources for photosynthetic CO_2_ assimilation [11]. The application of R and B LEDs for indoor cultivating basil has been investigated in many studies. Pennisi et al. (2019) [12] identified the role of the R: B ratio on the resource use efficiency of indoor basil cultivation, linking the physiological response to light with changes in yield and nutritional properties. The ratio of 3R:1B spectral fractions with a photosynthetic photon flux density (PPFD) of 215 ± 5 μmol·m^−2^·s^−1^ at a day/night cycle of 16/8 h provided high yield and chlorophyll content as well as improved water and energy use efficiency. It was concluded that the ratio of 3R:1B provided optimal growing conditions for the indoor cultivation of basil, fostering improved performances in terms of growth, physiological and metabolic functions, and resources use efficiency. Other research results reported that the ratio of 7R:3B in treatment with PPFD of 250 ± 10 μmol·m^−2^·s^−1^ and a day/night cycle of 16/8 h is optimal for basil [13]. In this light treatment, the highest antiradical activity, as well as the total phenolic and anthocyanin content were obtained. In particular, an amount of total phenolic content of 736.22 μg GAE/g FW that was achieved was 41 times higher compared to that obtained with white (W) light (17.86 μg GAE/g FW). The highest antiradical activity of 97.98% was also observed, much higher than the one (52.67%) recorded under R light. Essential oil compounds were shown to be strongly influenced by spectral light composition. It was reported that light spectra can coordinately change the composition of these compounds in the essential oil. It means that exposure to a specific light spectrum can alter the composition of the bioactive compounds in basil. Another study [14] Lobiuc et al. (2017) using different ratios of R: B illumination for basil showed that higher ratios of blue wavelength in the light treatments resulted in an increase in both chlorophyll a and chlorophyll b. Conversely, total phenolic and flavonoid contents, as well as free radical scavenging capacity, were higher when increasing the R light. At the ratio of 2R: 1B, the phenolic and flavonoid contents increase 1.87 and 2.06-fold, respectively, compared to W light.

Other less-efficient wavelengths such as far-red (FR), green (G), and ultraviolet (UV) lights are also important environmental signals for plants [15]. FR light reversed the effects of phytochrome leading to changes in gene expression, plant structure, and reproductive response [16]. FR-rich light increased stem length; enhanced apical dominance; accelerated flowering; reduced anabolic storage capacity, seed set, branching; shortened fruit development, and reduced seed quality [17,18]. G light has the lowest quantum yield, but it can penetrate deeper into the tree canopy due to its high transmittance and reflectance than other wavelengths, which explains why G light affects vegetative growth, organ growth, and plant tropism [16]. For basil, the addition of G light into the light treatment was reported by Lin et al. (2021) [19]. Results show that the ratio of 4R:1G:1B treatment provided many positive effects on the growth, development, and appearance of basil plants compared to 2:1:1 or 1:1:1 treatment. UV light is one important color that in many cases stimulates the production of secondary metabolites in plants [20], it also increases essential oils and enhances plant defense and protection against UV light [21]. UV-B or combined UV-A and UV-B light with white light were more effective than white light alone in increasing l-menthol and limonene concentrations in *Mentha arvensis* [22]. UV-A light treatment protects basil microgreens from hypocotyl elongation and enhances antioxidant properties in green-leaf basils increasing the significantly ABTS radical scavenging activity as well as higher ascorbic acid synthesis [23]. The addition of UV spectrum in the background RGB light (at the R: G: B ratio of 30.9:21.6:47.5) for basil treatment was reported by Kang et al. (2022) [24]. UV-A light (365–399 nm) at four dosages of 0, 35, 65 and 97 μmol·m^−2^·s^−1^ with the photoperiod of 16 h day/8 h night was added to the background RGB light for 14 days. It was reported that at the mild UV-A radiation (dosages of 35 and 65 μmol·m^−2^·s^−1^) supplemented with the main light, both yield and quality (the contents of acids, sugars, anthocyanins, and antioxidants) of sweet basil were increased.

The impact of mixed light based on R and B regions with UV-A, G and FR radiation for basil in terms of enhancing its composition, production, and antimicrobial activity of essential oil of basil planted in Vietnam has not been addressed yet. The rapid development of light-emitting diode (LED) technologies improves the economic output of plant production and quality. Therefore, the growth, composition, production, and antimicrobial activity of essential oils of basil planted in a net-house can be improved by supplemental light. In the present study, we focus on evaluating the effect of supplemental light sources in terms of spectral distribution and daily light dose (duration and illuminance) on the growth, composition, production, and antimicrobial activity of essential oils of basil planted in the net-house in Hanoi, Vietnam.

## 2. Results and Discussion

In this paper, we report on effects of different supplemental light spectra and doses on the essential oil productivity, composition and antimicrobial activity of basil cultivated in the net-house in Vietnam. The supplemental light conditions used for basil growing during four months were different in (i) ratio of light spectra: R:B:G:UVA:FR:W (that was presented in four groups of R:B:UVA~71:20:9, R:B:G~75:21:4, R:B:G:FR~43.5:43.5:8:5 and control with only natural sunlight, (ii) time of lighting: 4, 6, and 8 h/day, and (iii) intensity of light: 100, 150 and 200 μmol·m^−2^·s^−1^. The details of supplemental light conditions are presented later in the section “Materials and Methods” below.

### 2.1. The Effect of Light Spectra on Biomass and Essential Oil Production of O. basilicum

Under different supplemental light conditions, the biomass and essential oil yield of basil showed differences (Table 1). The highest plant height was found in the lighting formula F9 (118.55 cm/plant), followed by the one in the lighting formula F5 (104.50 cm/plant). These two values were significantly different (*p* = 0.001) with each other and with the ones in the other 8 formulas that ranged from 89.63 to 97.50 cm/plant. The formula F9 also gave the highest plant fresh weight (513.32 g/plant), followed by the formula F8 (390.65 g/plant) and F2 (399.50 g/plant). While the values varied from 290.40 to 334.62 g/plant from other formulas compared with 295.56 g/plant from the control formula F10.

The supplemental light conditions affected the water content of basil fresh biomass. The plant water content of two formulas F8 and F9 was similar to the control formula F10. However, significant reductions (*p* < 0.001) in the plant water content were observed in all other formulas that varied from 73.30 % (in F2) to 77.04 % (in F1). The essential oil yields of plants in the supplemental light formulas were significantly different (*p* < 0.001) ranging from 0.88 to 1.30% (*v*/*w*), calculated on a dry weight (DW) basis, compared with 1.21% (*v*/*w*) in the control formula F10. In line with data of plant height and biomass, the highest essential oil content was found in F9 (1.30%) that indicates the optimal supplemental light condition for basil growth and oil production. The increase in the essential oil content was also found in the formula F4 (1.26%) as well as formula F8 (1.25%).

The estimated production of basil in the supplemental light formulas F2 and F9 showed the significant higher values than the other formulas at 0.05 level (*p* < 0.001), namely respective productions of fresh biomass (15.58 ton/ha and 20.02 ton/ha), dry biomass (4.16 ton/ha and 3.55 ton/ha), and oil (44.03 L/ha and 46.11 L/ha).

Thus, supplemental light condition of formulas F2 (71% R: 20% B: 9% UVA at 120 µmol·m^−2^·s^−1^ in 6 h/day) and F9 (43.5% R:43.5% B:8.0% G:5.0% FR at 100 µmol·m^−2^·s^−1^ in 6 h/day) were the most suitable for the growth of basil that produced the highest biomass and essential oil productivity among all supplementary light conditions of the present study (Table 1).

The ranges of basil fresh biomass, dry biomass and oil production in this research are slightly higher than the ones previously reported [25]. It was reported that the fresh and/or dry weight of basil under different lighting conditions can be varying depending on the time of year [26]. Other research indicated a positive relationship between basil biomass and the Daily Light Integrals (DLIs) observed [27,28] when growing basil under DLI conditions from 7.0–17.8 mol·m^−2^·d^−1^ that is not clear in the present study with the range of total DLIs (sum of average PPFD value and daily supplemental light) from 11.21–15.53 mol·m^−2^·d^−1^.

### 2.2. The Effect of Light Spectra on Essential Oil Composition of O. basilicum

In addition to the oil yield, the quality of basil is also evaluated by the chemical composition of its essential oil. Hydrodistillation of the shredded aboveground biomass of basil harvested from ten formulas produced pale yellow oils with the average relative densities of the oils *d*^20^ ranged from 0.951 to 0.961 g/mL, the refractive indices *n*^20^ ranged from 1.509 to 1.512 and the equal optical rotations [α]D^20^ = [+] 0.26° (Table 2). These values of oil densities are roughly equal to the ones previously reported but the values of refractive indices are slightly higher [29].

The identification of compounds present in the basil essential oils harvested from ten formulas was carried out using mass spectral (MS) and retention index (RI) data. Table 3 presents the identified compounds in order of their elution on the HP-5MS column used for the GC-MS analysis.

A total of 20 to 27 compounds representing from 99.1% to 99.9% (by mass intensity) of the compositions were identified in the essential oils of basil cultivated under 10 different supplemental light conditions. These were comprised of dominant benzenoid aromatics ranging from 88.4 to 91.6% of the oils. While monoterpene hydrocarbons, oxygenated monoterpenes, sesquiterpene hydrocarbons, and oxygenated sesquiterpenes of the oils were at very low concentrations.

The difference between composition of the basil oils in 10 different supplemental lighting formulas is negligible with respect to the concentrations of methyl chavicol (=estragole); the main component ranged from 87.4 to 90.6%. The order of methyl chavicol content in basil oils is as follows: F7 = F8 > F10 > F2 = F5 > F3 > F4 = F9 > F6 > F1 (Table 3). Additional lighting time and light intensity may have influenced the concentration of basil essential oil compounds. Specifically, under the conditions of lighting time of 6 h/day (formulas F2 and F5) and light intensity of 150–200 µmol·m^−2^·s^−1^ (formulas F7 and F8), the methyl chavicol concentrations were higher than under the other conditions of lighting time and light intensity (Table 3).

The high content of methyl chavicol in basil in the formulas F2 and F5 could be influenced by both lighting duration and illuminance. Siposs et al. (2021) and Ahmed et al. (2020) [30,31] suggested that the optimal lighting condition for basil is 16–18 h/day at PPFD 200–250 µmol·m^−2^·s^−1^, which is comparable to the formulas F2 and F5 in the present study with supplemental lighting conditions of 6 h/day (total lighting duration of 20 h/day) at PPFD 120 µmol·m^−2^·s^−1^. In the formulas F7 and F8, the highest methyl chavicol content in basil can be attributed, in addition to lighting duration and illuminance, which is 6 h/day (total lighting duration is 20 h/day) at PPFD 150–200 µmol·m^−^^2^·s^−^^1^ -equivalent to DLI 14.45–15.53 mol·m^−2^·d^−1^, the composition of the light spectra has a higher ratio of B (43.5%). Its value is in agreement with the results reported by Hosseni (2018) [13] where the highest methyl chavicol was obtained under light B at DLI 14.4 mol·m^−2^·d^−1^ (16 h/day and PPFD 250 µmol·m^−2^·s^−1^).

Basil oil compositions have been divided into 7 chemotypes: (1) high linalool, (2) linalool/eugenol, (3) methyl chavicol without linalool, (4) methyl chavicol/linalool, (5) methyl eugenol/linalool, (6) methyl cinnamate/linalool and (7) bergamotene chemotypes [32]. Many earlier studies reported different major components identified in basil essential oils such as linalool, estragole and 1,8-cineole [33], 1,8-cineole, linalool, and geraniol [34], linalool followed by methyl eugenol and 1,8-cineol [35], methyl chavicol and methyl eugenol [36], linalool and methyl chavicol [7], methyl chavicol = estragole [37,38]. The differences in the main components of basil oil were mainly reported to be due to different varieties and geographical regions and countries. However, it is also revealed from the literature that light spectra affect the contents of some main compounds of basil essential oil [39], especially the content of methyl chavicol. Under condition of W light (intensity of 250 ± 10 µmol·m^−2^·s^−1^ PPFD, day/night cycles of 16/8 h after 30 days), methyl chavicol content in purple basil oil was highest (71.87%) [13]. The present study results are also similar to that of the published study. Specifically, the content of methyl chavicol in basil essential oil in formulas F7 and F8 supplemented with W light (additional lighting 3W: 5B: 6R: 1FR, at intensity of 150–200 µmol·m^−2^·s^−1^ reached the highest value (90.6%).

### 2.3. The Effect of Light Spectra on Antimicrobial Activity of Essential Oil of O. basilicum

The essential oils of basil grown in 10 different formulas were evaluated for antimicrobial activity against 7 strains: *Staphylococcus aureus*, *Bacillus subtilis*, *Lactobacillus fermentum*, *Salmonella enterica*, *Escherichia coli*, *Pseudomonas aeruginosa*, and *Candida albicans*. The essential oils of basil in 6 formulas from F2 to F7 showed weak inhibitory effects on *B. subtilis* with respective IC_50_ values of 6223, 6144, 12,698, 13,995, 5660, and 6349 µg/mL. The IC_50_ values of the oils on other 6 microorganism strains and all the 7 MIC values of the oils exceeded 16,384 µg/mL. Table 4 presents basil oil samples with inhibition activity against tested microbial strains.

The presence of high concentration of methyl chavicol and low concentration of linalool in the Basil oils in the present study may explain for the weak antimicrobial activity of the oils. A previous study showed that methyl chavicol is devoid of antimicrobial activity [40]. Likewise, basil oil with major constituents identified as methyl eugenol (39.3%) and methyl chavicol (38.3%) was found to be active against bacteria and fungi [36]. On the other hand, some other studies demonstrated that higher linalool-containing oils of basil exhibited higher antimicrobial activity [41,42].

No relationship was found between supplemental lighting conditions and the antimicrobial activity of basil essential oil in this study. However, previous studies showed that the antimicrobial activity of essential oils depends on their main compounds. On the other hand, supplemental lighting for basil with different light spectra can alter the content of its oil main compounds. Therefore, in the future, it is possible to find a certain light spectrum/mixture of light spectra that drastically changes the contents of the main compounds in basil essential oil, thereby increasing the oil’s antimicrobial activity.

## 3. Materials and Methods

### 3.1. Plant Materials and Growth Conditions

The experiments were set up in a net-house in Hanoi, VN (N 21°04′08′′, E 105°45′50′′) with 40% diffused light transmission being determined using relating photosynthetic photon flux density (PPFD) (400–700 nm) inside to outside the net-house. The basil seeds, purchased from Duc Thang Company in Hanoi, were sown in March 2021, then the seedlings were transplanted in April and a supplemental light experiment was held from May to September 2021. Daylight illuminance in a net-house was measured by the DigiSense Data Logging Light Meter (Model 20250-00), which was converted into Daily Light Integral (DLI) using conversion factors as described in the literature [43,44]. An average PPFD value of 11.21 mol·m^−2^·d^−1^ was determined for experimental time (in May 11.34, June: 11.44, July: 12.04, August: 10.92 and September: 10.31 mol·m^−2^·d^−1^). The supplemental LED lighting was applied after sunset and before sunrise with ten different light formulations as described in Table 5 with 15 plants in each formula and 3 replications. LED lights generated a wide continuous spectrum with UV (peak at 365 nm), B (peak at 440 nm), G (peak at 530 nm, R (peak at 660 nm), and FR (peak at 730 nm) measured by USB2000+ Fiber Optic Spectrometer, shown in Figure 1. The techniques for planting, caring, fertilizing, and harvesting basil plants were carried out according to the previous document [45].

### 3.2. Water Content Determination

Each basil sample was determined the water content using A&D Weighing AD-4714A General purpose moisture determination balance.

### 3.3. Essential Oil Isolation

Each basil sample consisting of aerial biomass of 15 plants (4.3–7.7 kg) was shredded and hydrodistilled for 4 h using a Clevenger type apparatus according to the previously published procedure [46] (Ministry of Health of Vietnam, 2017). The essential oil was then separated and stored at –5 °C for further analysis.

### 3.4. Determination of Essential Oil Physical Properties

Three physical properties of the essential oil including: relative density, refractive index and optical rotation were determined using the methods from the standards ISO 279:1998, ISO 280:1998 and ISO 592:1998, respectively.

### 3.5. GC-MS and GC-FID Analysis

The essential oils were analyzed by GC/MS-FID using an Agilent GC7890A system with Mass Selective Detector (Agilent 5975C). An HP-5MS fused silica capillary column (60 m × 0.25 mm i.d. × 0.25 μm film thickness) was used. Helium was the carrier gas with a flow rate of 1.0 mL/min. The inlet temperature was 250 °C and the oven temperature program was as follows: 60 °C to 240 °C at 4 °C/min. The split ratio was 1:100, the detector temperature was 270 °C, and the injection volume was 1 μL. The MS analysis was carried out at interface temperature 270 °C, MS mode, E.I. detector voltage 1258 V, and mass range 35–450 Da at 1.0 scan/s. FID analysis was carried out using the same chromatographic conditions. The FID temperature was 270 °C. Essential oil constituents were identified by their relative retention indices, determined by co-injection of a homologous series of *n*-alkanes (C_5_–C_30_), as well as by comparison of their mass spectral fragmentation patterns with those stored on the MS library NIST08, Wiley09, HPCH1607 [47,48] (Adams, 2017; Linstrom & Mallard, 2021). Data processing software was MassFinder 4.0 [49]. Component relative concentrations were calculated based on the area peak of FID chromatography without standardization.

### 3.6. Microbial Strains

The MIC and IC_50_ values of the oils were determined using 3 strains of Gram (+) bacteria including *Staphylococcus aureus* (ATCC 13709), *Bacillus subtilis* (ATCC 6633) and *Lactobacillus fermentum* (VTCC N4), 3 strains of Gram (−) bacteria including *Salmonella enterica* (VTCC), *Escherichia coli* (ATCC 25922) and *Pseudomonas aeruginosa* (ATCC 15442), 1 strain of yeast *Candida albicans* (ATCC 10231). The ATCC strains were obtained from American Type Culture Collection; the VTCC strains were obtained from Vietnam Type Culture Collection, Institute of Microbiology and Biotechnology, Vietnam National University, Ha Noi.

### 3.7. Screening of Antimicrobial Activity

MIC and IC_50_ of the essential oils were measured by the microdilution broth susceptibility assay [50,51]. Stock solutions of the oil were prepared in dimethylsulfoxide (DMSO). Dilution series were prepared from 16,384 μg/mL to 2 μg/mL (2^14^, 2^13^, 2^12^, 2^11^, 2^10^, 2^9^, 2^7^, 2^5^, 2^3^, 2^1^ μg/mL) in sterile distilled water in micro-test tubes, from where they were transferred to 96-well microtiter plates. Bacteria grown in double-strength Mueller-Hinton broth or double-strength tryptic soy broth, and fungi grown in double-strength Sabouraud dextrose broth were standardized to 5 × 10^5^ and 1 × 10^3^ CFU/mL, respectively. The last row, containing only the serial dilutions of sample without microorganisms, was used as a negative control. Sterile distilled water and medium served as a positive control. After incubation at 37 °C for 24 h, the MIC values were determined at the well with the lowest concentration of agents that completely inhibited the growth of microorganisms. The IC_50_ values were determined by the percentage of microorganisms inhibited growth based on the turbidity measurement data of EPOCH2C spectrophotometer (BioTeK Instruments, Inc., Highland Park Winooski, VT, USA) and Rawdata computer software (Intercity Business Park Mechelen Noord, Zone L Mechelen, 2800 Belgium) according to the following equations:$$\%\;\mathrm{Inhibition}=\frac{OD{\;}_{control\left(+\right)-OD_{\;Test\;agent}}}{OD{\;}_{control\left(+\right)-OD_{control\;\left(-\right)}}}\times 100\%$$
$$IC_{50}=High_{Conc\;}-\frac{\left(High_{Inh\%\;}-50\%)\;(High_{Conc}-Low_{Conc}\right)}{\left(High_{Inh\%}-Low_{Inh\%}\right)}$$
where: *OD*: optical density; control (+): only cells in medium without antimicrobial agent; test agent: corresponds to a known concentration of antimicrobial agent; control (−): culture medium without cells. *High_Conc_*/*Low_Conc_*: voncentration of test agent at high concentration/low concentration; *High_Inh%_*/*Low_Inh%_*: % inhibition at high concentration/% inhibition at low concentration.

Reference materials: Ampicillin for Gram (+) bacteria with IC_50_ and MIC values in the ranges of 0.02–3.62 µg/mL and of 0.125–32.0 µg/mL, Cefotaxime for Gram (−) bacteria with IC_50_ and MIC values in the range of 0.07–4.34 µg/mL and of 0.5–32.0 µg/mL, Nystatin for fungal strains with IC_50_ and MIC values of 1.32 µg/mL and 8.0 µg/mL.

### 3.8. Statistical Analysis

The measurements of physiological parameters and essential oil contents of basil were analyzed by a single factor completely randomized analysis of variance (ANOVA), which compared the lighting treatments. For the significant values, means were separated by the least significant difference (LSD) test at *p* ≤ 0.05 using IRRISTAT ver. 5.0 (International Rice Research Institute, Los Baños, Philippines).

## 4. Conclusions

This is the first study to provide the information on the various aspects of basil cultivated under ten different supplemental light spectra and dosages in Vietnam. The average yields of basil essential oils from 0.88 to 1.30% (*v*/*w*), calculated on a DW basis, were obtained. Methyl chavicol (=estragole) was the unique main component of the oils ranging from 87.4 to 90.6%. Additional lighting gave out effects on the growth and essential oil content of basil, as shown by the significant differences (*p* < 0.001) in biomass, oil yield and oil production between the lighting formulas. Supplemental light conditions of formulas F2 (71% R:20% B:9.0% UVA at 120 µmol·m^−2^·s^−1^ in 6 h/day) and F9 (43.5% R:43.5% B:8.0% G:5.0% FR at 100 µmol·m^−2^·s^−1^ in 6 h/day) were the most suitable for the growth of basil that produced the highest biomass (fresh: 15.58 ton/ha and 20.02 ton/ha; dry: 4.16 ton/ha and 3.55 ton/ha) and essential oil productivities (44.03 L/ha and 46.11 L/ha) among all supplementary light conditions of the present study. Additional lighting time and light intensity may have influenced the concentration of the major component of basil essential oil with higher contents of methyl chavicol under conditions of 6 h/day (formulas F2 and F5) and 150–200 µmol·m^−2^·s^−1^ (formulas F7 and F8) than under the other conditions. The essential oils from basil in formulas from F2 to F7 showed weak inhibitory effects on only *Bacillus subtilis*. However, no effect of supplemental lighting conditions on the antimicrobial activity of basil essential oil was recorded. The results of present study can be the basis for future research to promote the increase in biomass yield, production and quality of essential oils of plants.

## Figures and Tables

**Figure 1 molecules-27-05599-f001:**
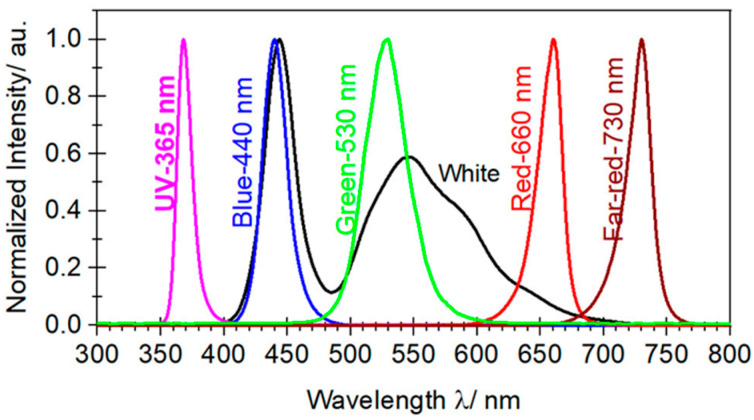
Normalized spectral distribution of supplemental LEDs lighting.

**Table 1 molecules-27-05599-t001:** Biomass and essential oil production of *O. basilicum* cultivated under different light conditions.

Formula	Height of Plant (cm/plant)	Fresh Weight of Plant (g/plant)	Production of Fresh Biomass (ton/ha) *	Water Content (%)	Production of Dry Biomass (ton/ha) *	Essential Oil Content (%, Dry *v*/*w*)	Production of Essential Oil (L/ha) *
F1	94.39 ^a^ ± 9.7	290.40 ^a^ ± 8.2	11.33 ^a^ ± 0.3	77.04 ^b^ ± 2.9	2.60 ^b^ ± 0.07	1.02 ^b^ ± 0.02	26.64 ^a^ ± 0.8
F2	94.03 ^a^ ± 5.0	399.50 ^c^ ± 14.3	15.58 ^b^ ± 0.6	73.30 ^a^ ± 0.3	4.16 ^f^ ± 0.15	1.06 ^c^ ± 0.01	44.03 ^e^ ± 1.6
F3	94.87 ^a^ ± 3.2	325.87 ^a^ ± 12.6	12.71 ^a^ ± 0.5	74.27 ^a^ ± 0.3	3.27 ^d^ ± 0.13	1.21 ^d^ ± 0.02	39.54 ^d^ ± 1.5
F4	97.50 ^a^ ± 5.3	332.07 ^b^ ± 10.8	12.95 ^b^ ± 0.4	75.00 ^a^ ± 0.9	3.24 ^d^ ± 0.11	1.26 ^f^ ± 0.02	40.87 ^d^ ± 1.3
F5	104.50 ^b^ ± 11.1	334.62 ^b^ ± 30.5	13.05 ^b^ ± 1.2	76.00 ^a^ ± 1.0	3.13 ^d^ ± 0.29	1.22 ^e^ ± 0.02	38.30 ^d^ ± 3.5
F6	89.63 ^a^ ± 3.6	306.73 ^a^ ± 7.9	11.96 ^a^ ± 0.3	74.72 ^a^ ± 1.9	3.02 ^c^ ± 0.08	0.88 ^a^ ± 0.02	26.56 ^a^ ± 0.7
F7	92.20 ^a^ ± 5.5	308.12 ^a^ ± 5.4	12.02 ^a^ ± 0.2	74.40 ^a^ ± 1.0	3.08 ^c^ ± 0.05	1.07 ^c^ ± 0.03	32.89 ^b^ ± 0.6
F8	96.22 ^a^ ± 4.9	390.65 ^c^ ± 21.2	15.24 ^b^ ± 0.8	80.99 ^c^ ± 3.1	2.90 ^c^ ± 0.16	1.25 ^e^ ± 0.03	36.12 ^c^ ± 2.0
F9	118.55 ^c^ ± 6.3	513.32 ^d^ ± 41.9	20.02 ^c^ ± 1.6	82.26 ^c^ ± 3.2	3.55 ^e^ ± 0.29	1.30^g^ ± 0.02	46.11 ^f^ ± 3.8
F10	90.93 ^a^ ± 9.7	295.56 ^a^ ± 33.2	11.53 ^a^ ± 1.3	81.08 ^c^ ± 3.5	2.18 ^a^ ± 0.24	1.21 ^d^ ± 0.02	26.35 ^a^ ± 3.0
5%LSD	10.176	37.935	1.479	3.695	0.304	0.034	3.715

Note: Mean values followed by the same letter within a column are not statistically different for 0.05 significant level, * Estimated values (cultivation distance of basil: 40 × 50 cm, estimated number: 39,000 plants per ha).

**Table 2 molecules-27-05599-t002:** Some physical properties of essential oil of *O. basilicum* cultivated under different light conditions.

Parameters	F1	F2	F3	F4	F5	F6	F7	F8	F9	F10
Relative density *d*^20^	0.953	0.951	0.955	0.951	0.954	0.960	0.960	0.961	0.951	0.958
Refractive index *n*^20^	1.509	1.511	1.510	1.510	1.511	1.510	1.511	1.512	1.509	1.511
Optical rotation [α]_D_^20^	[+]0.26	[+]0.26	[+]0.26	[+]0.26	[+]0.26	[+]0.26	[+]0.26	[+]0.26	[+]0.26	[+]0.26

**Table 3 molecules-27-05599-t003:** Composition of essential oils of *O. basilicum* cultivated under different light conditions.

N°	RI	Compounds	F1	F2	F3	F4	F5	F6	F7	F8	F9	F10
1	850	(3*Z*)-Hexen-1-ol	0.4	0.2	0.3	0.3	0.5	0.3	0.3	0.3	0.6	0.5
2	939	*α*-Pinene	0.1	-	0.1	-	-	-	-	-	-	-
3	976	1-Octen-3-ol	0.2	-	-	-	0.2	-	-	-	-	-
4	984	*β*-Pinene	0.1	-	0.1	0.1	0.1	0.1	-	-	-	-
5	991	*β*-Myrcene	0.7	0.6	0.5	0.5	0.5	0.5	0.4	0.4	0.5	0.3
6	1033	Limonene	0.3	0.2	0.2	0.2	0.2	0.2	0.2	0.2	0.2	0.2
7	1037	1,8-Cineole	0.8	0.7	0.9	0.7	0.7	0.7	0.7	0.7	0.8	0.7
8	1048	(*E*)-*β*-Ocimene	1.9	1.7	1.9	2.0	1.4	1.5	1.3	1.3	1.9	1.1
9	1096	Fenchone	0.1	-	-	-	-	-	-	-	0.1	-
10	1101	Linalool	0.6	0.5	0.7	0.5	0.5	0.4	0.6	0.6	0.8	0.7
11	1121	*endo*-Fenchol	0.3	0.3	0.3	0.2	0.2	0.3	0.3	0.3	0.3	0.3
12	1155	Camphor	0.4	0.2	0.4	0.4	0.2	0.3	0.2	0.2	0.4	0.2
13	1175	Borneol (=*endo*-Borneol)	0.2	0.2	0.2	0.2	0.3	0.2	0.2	0.2	0.2	0.3
14	1198	*α*-Terpineol	0.3	0.2	0.2	0.2	0.2	0.2	0.2	0.2	0.2	0.2
15	1207	Methyl chavicol (=Estragole)	87.4	89.7	88.8	88.6	89.7	88.4	90.6	90.6	88.6	89.8
16	1228	Fenchyl acetate	0.4	0.4	0.3	0.3	0.2	0.4	0.3	0.3	0.2	0.3
17	1253	Chavicol	-	-	-	-	0.1	0.1	-	0.1	-	-
18	1294	Bornyl acetate	0.5	0.5	0.4	0.4	0.4	0.4	0.5	0.5	0.3	0.5
19	1403	*cis*-*β*-Elemene	0.2	0.2	0.2	0.3	0.2	0.2	0.1	0.2	0.3	0.1
20	1407	Methyleugenol	1.0	1.0	0.9	0.9	0.8	1.3	1.0	0.9	0.8	1.2
21	1437	(*E*)-*β*-Caryophyllene	-	0.2	-	0.1	0.2	0.2	0.1	-	0.2	0.1
22	1446	*trans*-*α*-Bergamotene	1.8	1.6	1.6	1.5	1.2	1.6	1.5	1.5	1.3	1.3
23	1452	*α*-Guaiene	0.1	-	0.1	0.1	0.1	0.1	-	-	0.1	-
24	1471	*α*-Humulene	-	-	0.1	0.1	0.1	0.1	-	-	-	-
25	1498	Germacrene D	0.1	0.2	0.2	0.3	0.2	0.2	0.1	0.1	0.2	0.1
26	1513	(*Z*)-*α*-Bisabolene	0.2	-	-	0.2	-	-	-	-	-	-
27	1521	*α*-Bulnesene (=*δ*-Guaiene)	0.1	0.1	0.2	0.2	0.2	0.1	0.1	-	0.2	0.1
28	1530	*γ*-Cadinene	0.3	0.3	0.3	0.3	0.3	0.3	0.3	0.3	0.3	0.3
29	1634	1,10-di-*epi*-Cubenol	0.1	0.1	0.1	0.1	0.1	0.2	0.1	-	0.1	0.1
30	1658	*epi*-*α*-Cadinol (=*τ*-Cadinol)	0.7	0.8	0.8	0.8	0.8	1.0	0.7	0.7	0.9	0.7
Total			99.3	99.9	99.8	99.5	99.6	99.3	99.8	99.6	99.5	99.1
Number of compounds identified			27	22	25	26	27	26	22	20	24	22
Monoterpene hydrocarbons			3.1	2.5	2.8	2.8	2.2	2.3	1.9	1.9	2.6	1.6
Oxygenated monoterpenes			3.6	3.0	3.4	2.9	2.7	2.9	3.0	3.0	3.3	3.2
Sesquiterpene hydrocarbons			2.8	2.6	2.7	3.1	2.5	2.8	2.2	2.1	2.6	2.0
Oxygenated sesquiterpenes			0.8	0.9	0.9	0.9	0.9	1.2	0.8	0.7	1.0	0.8
Others			0.6	0.2	0.3	0.3	0.7	0.3	0.3	0.3	0.6	0.5
Benzenoids			88.4	90.7	89.7	89.5	90.6	89.8	91.6	91.6	89.4	91.0

**Table 4 molecules-27-05599-t004:** Antimicrobial activity of essential oils of *O. basilicum* cultivated under different light conditions.

Formula	Values (µg/mL)	Antimicrobial Activity of *O. basilicum* Essential Oils						
		Gram (+) Bacteria			Gram (–) Bacteria			Yeast
		*Staphylococcus aureus*	*Bacillus subtilis*	*Lactobacillus fermentum*	*Salmonella enterica*	*Escherichia coli*	*Pseudomonas* *aeruginosa*	*Candida albicans*
F1	IC_50_	>16,384	>16,384	>16,384	>16,384	>16,384	>16,384	>16,384
	MIC	>16,384	>16,384	>16,384	>16,384	>16,384	>16,384	>16,384
F2	IC_50_	>16,384	6223	>16,384	>16,384	>16,384	>16,384	>16,384
	MIC	>16,384	>16,384	>16,384	>16,384	>16,384	>16,384	>16,384
F3	IC_50_	>16,384	6144	>16,384	>16,384	>16,384	>16,384	>16,384
	MIC	>16,384	>16,384	>16,384	>16,384	>16,384	>16,384	>16,384
F4	IC_50_	>16,384	12,698	>16,384	>16,384	>16,384	>16,384	>16,384
	MIC	>16,384	>16,384	>16,384	>16,384	>16,384	>16,384	>16,384
F5	IC_50_	>16,384	13,995	>16,384	>16,384	>16,384	>16,384	>16,384
	MIC	>16,384	>16,384	>16,384	>16,384	>16,384	>16,384	>16,384
F6	IC_50_	>16,384	5660	>16,384	>16,384	>16,384	>16,384	>16,384
	MIC	>16,384	>16,384	>16,384	>16,384	>16,384	>16,384	>16,384
F7	IC_50_	>16,384	6349	>16,384	>16,384	>16,384	>16,384	>16,384
	MIC	>16,384	>16,384	>16,384	>16,384	>16,384	>16,384	>16,384
F8	IC_50_	>16,384	>16,384	>16,384	>16,384	>16,384	>16,384	>16,384
	MIC	>16,384	>16,384	>16,384	>16,384	>16,384	>16,384	>16,384
F9	IC_50_	>16,384	>16,384	>16,384	>16,384	>16,384	>16,384	>16,384
	MIC	>16,384	>16,384	>16,384	>16,384	>16,384	>16,384	>16,384
F10	IC_50_	>16,384	>16,384	>16,384	>16,384	>16,384	>16,384	>16,384
	MIC	>16,384	>16,384	>16,384	>16,384	>16,384	>16,384	>16,384

**Table 5 molecules-27-05599-t005:** Supplemental light conditions in cultivation of *O. basilicum*.

Formulas	Spectral Distribution	Duration (h/day)	Lighting Time	Light Intensity (µmol·m^−2^·s^−1^)	Total Daily Supplemental Light (mol·m^−2^·d^−1^)	Total Daily Light (mol·m^−2^·d^−1^)
F1	R:B:UVA~71:20:9	8	1:00–5:00 am and 19:00–23:00 pm	120 ± 10	3.456	14.666
F2		6	2:00–5:00 am and 19:00–22:00 pm	120 ± 10	2.592	13.802
F3		4	3:00–5:00 am and 19:00–21:00 pm	120 ± 10	1.728	12.938
F4	R:B:G ~75:21:4	8	1:00–5:00 am and 19:00–23:00 pm	120 ± 10	3.456	14.666
F5		6	2:00–5:00 am and 19:00–22:00 pm	120 ± 10	2.592	13.802
F6		4	3:00–5:00 am and 19:00–21:00 pm	120 ± 10	1.728	12.938
F7	R:B:G:FR~43.5:43.5:8:5	6	2:00–5:00 am and 19:00–22:00 pm	200 ± 10	4.32	15.53
F8		6	2:00–5:00 am and 19:00–22:00 pm	150 ± 10	3.24	14.45
F9		6	2:00–5:00 am and 19:00–22:00 pm	100 ± 10	2.16	13.37
F10	Control	0	0	0	0	11.21

## Data Availability

All data are available in this publication.

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
