# Peer review of "Effects of Supplemental Light Spectra on the Composition, Production and Antimicrobial Activity of Ocimum basilicum L. Essential Oil"

_molecules, 2022, doi:10.3390/molecules27175599_

Round 1

Reviewer 1 Report

Comments to the Authors.

The manuscript entitled “Effects of Supplemental Light Spectra on the Composition, Production and Antimicrobial Activity of Ocimum basilicum L. Essential Oil.” has been reviewed. The manuscript is interesting and complete, I suggest some very minor revisions.

A minor revision of the language is suggested.

Line 58 – “CO2“ instead of “CO2”.

Line 99 and 100 – Menta arvensis must be in italics

Line 269 – Bacillus subtilis must be in italics

Due to the above-mentioned correction, I suggest to accept the MS after minor revision (corrections to minor methodological errors and text editing).

Author Response

Dear Editorial board of Journal “Molecules”,

Dear Prof. Dr. Wen-Ling Shih - Guest Editor of Special Issue "Essential Oil Research and Product Development", section "Natural Products Chemistry",

Dear the Reviewer 1, 

thank you for the comments from the reviewer for our manuscript entitled “Effects of Supplemental Light Spectra on the Composition, Production and Antimicrobial Activity of Ocimum basilicum L. Essential Oil”, molecules-1871115. We would like to send you the revised manuscript with all changes marked up using the “Track Changes” function.

We changed the name order of one of our co-authors – Assoc. Prof. Dr. Tran Quoc Tien into right order in Vietnamese mane and added his Orcid number at his desire.

English language was checked by one of our co-authors – Prof. Dr. William N. Setzer – a native English-speaking colleague.

In correcting the manuscript, we have addressed the concerns of the Reviewer 1 as follows:

Reviewer 1: A minor revision of the language is suggested.

Line 58 – “CO2“ instead of “CO2”.

Line 99 and 100 – Mentha arvensis must be in italics

Line 269 – Bacillus subtilis must be in italics

Response: Thank you very much for the remark on the formatting errors. All of those mistake have been revised into correct format (new line numbers: 58, 99-100 and 368).

Thank you for your consideration of this manuscript.

Sincerely yours,

Ha Thi Thu CHU

Reviewer 2 Report

In this study, the effects of different supplemental light spectra and doses (duration and illumination) on the essential oil of basil (Ocimum basilicum L.) cultivated in Vietnamese net chambers over a period of four months were investigated. The research is interesting and important. The relationship between light treatment conditions and chemical composition and activity should be explained before publication.

1. The data in Table 1 should be presented in the form of mean ± SD.

2. Where did the Production of fresh biomass (ton/ha) data in Table 1 come from? In lines 242-245, it is stated that with 15 plants in each formula and 3 replications, and these amounts do not amount to as much as 1 ha planted.

3.The data in Production of essential oil (L/ha) in Table 1 needs to be determined again.

4.The method of determining the water content should be mentioned in the Materials and Methods.

5. Lines 211-212. “The IC50 values of the oils on other 6 microorganism strains and all the 7 MIC values of the oils exceeded 16.384 µg/mL.” In Table 4 it shows that the essential oil had no inhibitory effect on the other 6 strains, while the IC50 values of 16.384 µg/mL written here indicates a good inhibitory effect.

6. There are some writing formatting errors in the manuscript, such as CO2 in line 58; IC50 in 4.5 and 4.6.

7. In 4.6 it is mentioned that MIC values were measured, however, in the results of 2.3 there is no MIC value.

8. No relationship was found between supplemental lighting conditions and the antimicrobial activity of basil essential oil in this study. The aim of changing the production conditions is not only to increase the yield but also, above all, to increase the biological activity. It is mentioned in the introduction that UV-A light treatment can enhance antioxidant properties, why not do antioxidant experiments?

Author Response

Dear Editorial board of Journal “Molecules”,

Dear Prof. Dr. Wen-Ling Shih - Guest Editor of Special Issue "Essential Oil Research and Product Development", section "Natural Products Chemistry",

Dear the Reviewer 2, 

thank you for the comments from the reviewer for our manuscript entitled “Effects of Supplemental Light Spectra on the Composition, Production and Antimicrobial Activity of Ocimum basilicum L. Essential Oil”, molecules-1871115. We would like to send you the revised manuscript with all changes marked up using the “Track Changes” function.

We changed the name order of one of our co-authors – Assoc. Prof. Dr. Tran Quoc Tien into right order in Vietnamese mane and added his Orcid number at his desire.

English language was checked by one of our co-authors – Prof. Dr. William N. Setzer – a native English-speaking colleague.

In correcting the manuscript, we have addressed the concerns of the Reviewer 2 as follows:

Reviewer 2: The relationship between light treatment conditions and chemical composition and activity should be explained before publication.

Response: Thank you very much for the comment on the manuscript. I have added some explanation based on your recommendation as follows:

At the bottom of page 6, one paragraph has been added to explain the relationship between light treatment conditions and basil essential oil chemical composition: “The high content of methyl chavicol in basil in the formulas F2 and F5 could be influenced by both lighting duration and illuminance. Siposs et al. (2021) and Ahmed et al. (2020) [30,31] suggested that the optimal lighting condition for basil is 16-18 hours/day at PPFD 200-250 µmol·m-2·s-1, which is comparable to the formulas F2 and F5 in the present study with supplemental lighting conditions of 6 h/day (total lighting duration of 20 h/day) at PPFD 120 µmol·m-2·s-1. In the formulas F7 and F8, the highest methyl chavicol content in basil can be attributed, in addition to lighting duration and illuminance, which is 6 hours/day (total lighting duration is 20 hours/day) at PPFD 150-200 µmol·m-2·s-1  - equivalent to DLI 14.45-15.53 mol·m-2·d-1, the composition of the light spectra has a higher ratio of B light (43.5%). Its value is in agreement with the results reported by Hosseni (2018) [13] where the highest concentration of methyl chavicol was obtained under B light at DLI 14.4 mol·m-2·d-1 (16 h/day and PPFD 250 µmol·m-2·s-1).”

In the previous research, Hosseni (2018) [13] reported that the highest concentration (89.84%) of methyl chavicol was obtained under condition of B light, followed by 81.55% under 50R:50B, and  73.77% under 70R:30B.

In addition, one more colunm has been inserted into Table 5 to present values of total light including natural sunlight and supplemental light for 10 lighting formulas.

At the bottom of page 7, we commented that: “No relationship was found between supplemental lighting conditions and the antimicrobial activity of basil essential oil in this study”.

Point 1: The data in Table 1 should be presented in the form of mean ± SD.

Response: The SD values have been added into Table 1.

Point 2: Where did the “Production of fresh biomass (ton/ha)” data in Table 1 come from? In lines 242-245, it is stated that with 15 plants in each formula and 3 replications, and these amounts do not amount to as much as 1 ha planted.

Response: The values of “Production of fresh biomass (ton/ha)” were estimated based on the cultivating method applied in this study as follows: with the planting distance between the basil plants of 40 x 50 cm, the number of basil plants cultivated per ha is 39,000 plants/ha.

Some more information has been added in the Note under Table 1: *Estimated values (cultivation distance of basil: 40 × 50 cm, estimated number: 39,000 plants per ha).

Point 3: The data in Production of essential oil (L/ha) in Table 1 needs to be determined again

Response: The data in Production of essential oil (L/ha) in Table 1 were calculated as follows:

Point 4: The method of determining the water content should be mentioned in the Materials and Methods

Response: We have added the method of determining the water content as you recommended.

Point 5: Lines 211-212. “The IC50 values of the oils on other 6 microorganism strains and all the 7 MIC values of the oils exceeded 16.384 µg/mL.” In Table 4 it shows that the essential oil had no inhibitory effect on the other 6 strains, while the IC50 values of 16.384 µg/mL written here indicates a good inhibitory effect.

Response: In this study, the dilution series were prepared from 16,384 μg/mL to 2 μg/mL (214, 213, 212, 211, 210, 29, 27, 25, 23, 21 μg/mL) when we screened the antimicrobial activity of basil essential oil samples against 7 strains of microorganisms. The results showed that the essential oils of basil in 6 formulas from F2 to F7 showed weak inhibitory effects on B. subtilis with respective IC50 values of 6223, 6144, 12,698, 13,995, 5660, and 6349 µg/mL. The IC50 values of the oils on other 6 microorganism strains and all the 7 MIC values of the oils exceeded 16,384 µg/mL.

Thank you for your comment. We have added the real values of IC50 and MIC into Table 4 as well as corrected the commas used to separate thousands (previous mistake: decimal point).

Point 6: There are some writing formatting errors in the manuscript, such as CO2 in line 58; IC50 in 4.5 and 4.6.

Response: All of those mistakes were revised into correct format.

Point 7: In 4.6 it is mentioned that MIC values were measured, however, in the results of 2.3 there is no MIC value.

Response: Thank you for your comment. The real values of IC50 and MIC have been added into Table 4.

Point 8: No relationship was found between supplemental lighting conditions and the antimicrobial activity of basil essential oil in this study. The aim of changing the production conditions is not only to increase the yield but also, above all, to increase the biological activity. It is mentioned in the introduction that UV-A light treatment can enhance antioxidant properties, why not do antioxidant experiments?

Response: This is a very interesting recommendation for us. We noted this issue for future research project. In this project, we harvested basil samples and distilled the essential oils right after the end of the planting experiment in 2021 and currently we are not able to do antioxidant activity testing.

Thank you for your consideration of this manuscript.

Sincerely yours,

Ha Thi Thu CHU

Reviewer 3 Report

The research manuscript entitled, “Effects of Supplemental Light Spectra on the Composition, 1 Production and Antimicrobial Activity of Ocimum basilicum L. 2 Essential Oil” is fascinating and informative research work. This paper may be of interest to Journal readers as well as other researchers working on this topic around the world. This work may be accepted for publication after minor revisions.

  1. Arrange keywords alphabetically.
  2. Follow journal reference pattern for references.
  3. Provide DOI numbers for all possible cited references.
  4. Include recent references in support of your research.

5.      Complete page numbers of the following references:

8. A. R., Mohebbati, R., & Boskabady, M. H. The effect of Ocimum basilicum L. and its main ingredients on respiratory dis-307 orders: An experimental, preclinical, and clinical review. Front. Pharmacol, 2022, 12.

15. Dou, H., Niu, G., Gu, M., & Masabni, J. G. Effects of light quality on growth and phytonutrient accumulation of herbs 325 under controlled environments. Horticulturae, 2017, 3(2), 36.

Author Response

Dear Editorial board of Journal “Molecules”,

Dear Prof. Dr. Wen-Ling Shih - Guest Editor of Special Issue "Essential Oil Research and Product Development", section "Natural Products Chemistry",

Dear the Reviewer 3,  

thank you for the comments from the reviewer for our manuscript entitled “Effects of Supplemental Light Spectra on the Composition, Production and Antimicrobial Activity of Ocimum basilicum L. Essential Oil”, molecules-1871115. We would like to send you the revised manuscript with all changes marked up using the “Track Changes” function.

We changed the name order of one of our co-authors – Assoc. Prof. Dr. Tran Quoc Tien into right order in Vietnamese mane and added his Orcid number at his desire.

English language was checked by one of our co-authors – Prof. Dr. William N. Setzer – a native English-speaking colleague.

In correcting the manuscript, we have addressed the concerns of the Reviewer 3 as follows:

Reviewer 3: This work may be accepted for publication after minor revisions.

Point 1: Arrange keywords alphabetically.

Response: Thank you for your comment. The keywords have been rearranged alphabetically.

Point 2: Follow journal reference pattern for references.

Response: We have followed journal reference pattern for references as your comment.

Point 3: Provide DOI numbers for all possible cited references.

Response: The DOI numbers have been added for all possible cited references.

Point 4: Include recent references in support of your research.

Response: We have added some more references [30,31] in support of explanation for our research results.

Point 5: Complete page numbers of the following references:

- 8. A. R., Mohebbati, R., & Boskabady, M. H. The effect of Ocimum basilicum L. and its main ingredients on respiratory dis-307 orders: An experimental, preclinical, and clinical review. Front. Pharmacol2022, 12.

- 15. Dou, H., Niu, G., Gu, M., & Masabni, J. G. Effects of light quality on growth and phytonutrient accumulation of herbs 325 under controlled environments. Horticulturae, 2017, 3(2), 36.

Response: We have added page numbers of the above references as follows:

- 8. Aminian, A. R., Mohebbati, R., & Boskabady, M. H., The effect of Ocimum basilicum L. and its main ingredients on respiratory disorders: An experimental, preclinical, and clinical review. Front Pharmacol, 2022, 3; 12, 805391. doi: 10.3389/fphar.2021.805391.

- 15. Dou, H., Niu, G., Gu, M., & Masabni, J. G. Effects of light quality on growth and phytonutrient accumulation of herbs under controlled environments. Horticulturae 2017, 3, 36; 11 pages, doi:10.3390/horticulturae3020036.

Thank you for your consideration of this manuscript.

Sincerely yours,

Ha Thi Thu CHU

Reviewer 4 Report

The report by Chu et al. is a straightforward work and while the results are not outstanding, they may form the basis for future exploration. The presentation is fair, could be better, and there is much copyediting to be done. I recommend publication with attention to the following points.

11.     Define DW.

22.     Define DLI at first usage.

33.     Define PPFD only once.

44.     The title to Table 5 is wrong. This table is so important that perhaps it should be placed at the very beginning of the Results and Discussion ? or at least explicit mention of it’s existence be made in the text.

55.     On top of page 6 (bottom section of Table 3), make it clear what the numbers are, % by weight, % by mass intensity or whatever it is.

Author Response

Dear Editorial board of Journal “Molecules”,

Dear Prof. Dr. Wen-Ling Shih - Guest Editor of Special Issue "Essential Oil Research and Product Development", section "Natural Products Chemistry",

Dear the Reviewer 4,  

thank you for the comments from the reviewer for our manuscript entitled “Effects of Supplemental Light Spectra on the Composition, Production and Antimicrobial Activity of Ocimum basilicum L. Essential Oil”, molecules-1871115. We would like to send you the revised manuscript with all changes marked up using the “Track Changes” function.

We changed the name order of one of our co-authors – Assoc. Prof. Dr. Tran Quoc Tien into right order in Vietnamese mane and added his Orcid number at his desire.

English language was checked by one of our co-authors – Prof. Dr. William N. Setzer – a native English-speaking colleague.

In correcting the manuscript, we have addressed the concerns of the Reviewer 4 as follows:

Reviewer 4: The presentation is fair, could be better, and there is much copyediting to be done. Moderate English changes required.

Response: Thank you for your comment. The manuscript has been proofread by our native English-speaking colleague.

Point 1: Define DW.

Response: DW has been defined in line 147.

Point 2: Define DLI at first usage.

Response: DLI has been defined at first usage in line 165.

Point 3: Define PPFD only once.

Response: PPFD definition was retained only once in line 63 (it was removed out of line 68).

Point 4: The title to Table 5 is wrong. This table is so important that perhaps it should be placed at the very beginning of the Results and Discussion ? or at least explicit mention of it’s existence be made in the text.

Response: The title to Table 5 has been corrected. We has mentioned on its existence in text at the very beginning of the Results and Discussion.

Point 5: On top of page 6 (bottom section of Table 3), make it clear what the numbers are, % by weight, % by mass intensity or whatever it is.

Response: The phrases “by mass intensity” and “of the oils” have been added in the first paragraph of page 6 to make the sentences more clearly.

Thank you for your consideration of this manuscript.

Sincerely yours,

Ha Thi Thu CHU

Round 2

Reviewer 2 Report

The author has revised the manuscript well enough to accept the manuscript in its present form.